# Saliva sampling method influences oral microbiome composition and taxa distribution associated with oral diseases

**Cristian Roca**[1,2‡], **Alaa A. Alkhateeb**[3,4‡], **Bryson K. Deanhardt**[5], **Jade K. Macdonald**[2], **Donald L. Chi**[4,6], **Jeremy R. Wang**[5], **Matthew C. Wolfgang**[1,2]*

**1** Department of Microbiology and Immunology, University of North Carolina at Chapel Hill, Chapel Hill, North Carolina, United States of America, **2** Marsico Lung Institute, University of North Carolina at Chapel Hill, Chapel Hill, North Carolina, United States of America, **3** Department of Dental Health Sciences, School of Applied Medical Sciences, King Saud University, Riyadh, Saudi Arabia, **4** Department of Oral Health Sciences, School of Dentistry, University of Washington, Seattle, Washington, United States of America, **5** Department of Genetics, University of North Carolina at Chapel Hill, Chapel Hill, North Carolina, United States of America, **6** Department of Health Systems and Population Health, School of Public Health, University of Washington, Seattle, Washington, United States of America

‡ CR and AAA are contributed equally to this work and are co-first authors on this work.
* matthew_wolfgang@med.unc.edu

## Abstract

Saliva is a readily accessible and inexpensive biological specimen that enables investigation of the oral microbiome, which can serve as a biomarker of oral and systemic health. There are two routine approaches to collect saliva, stimulated and unstimulated; however, there is no consensus on how sampling method influences oral microbiome metrics. In this study, we analyzed paired saliva samples (unstimulated and stimulated) from 88 individuals, aged 7–18 years. Using 16S rRNA gene sequencing, we investigated the differences in bacterial microbiome composition between sample types and determined how sampling method affects the distribution of taxa associated with untreated dental caries and gingivitis. Our analyses indicated significant differences in microbiome composition between the sample types. Both sampling methods were able to detect significant differences in microbiome composition between healthy subjects and subjects with untreated caries. However, only stimulated saliva revealed a significant association between microbiome diversity and composition in individuals with diagnosed gingivitis. Furthermore, taxa previously associated with dental caries and gingivitis were preferentially enriched in individuals with each respective disease only in stimulated saliva. Our study suggests that stimulated saliva provides a more nuanced readout of microbiome composition and taxa distribution associated with untreated dental caries and gingivitis compared to unstimulated saliva.

## Introduction

Worldwide, approximately 3.5 billion people are affected by oral diseases like dental caries and periodontitis, with an estimated direct cost for treatment approaching 300 billion USD

**Data Availability Statement:** All bacterial sequencing data is publicly available in fastq file

format at: NCBI Sequence Read Archive (accession code PRJNA1072698). Metadata for individual samples is publicly available at the Carolina Digital Repository (https://cdr.lib.unc.edu/) under the name of "Saliva Stimulation and microbiome project".

**Funding:** C.R., D.L.C and M.C.W. were funded in part by support from the National Institute of Dental and Craniofacial Research, NIH (U01DE030418), and J.R.W. was funded by NIH K01 DK119582. The funders had no role in study design, data collection and analysis, decision to publish, or preparation of the manuscript.

**Competing interests:** The authors have declared that no competing interests exist.

annually [1, 2]. Oral diseases have a disproportionate impact on low-resourced populations, leading to negative health outcomes including pain, sepsis, poor quality of life, and premature death [1, 3]. Early diagnosis and cost-effective interventions are needed to prevent life-threatening complications caused by oral diseases. An important aspect of developing such interventions is understanding microbiological mechanisms underlying prevention efforts [4].

Oral diseases like caries and gingivitis are mediated by bacterial dysbiosis. The formation of caries is primarily associated with excessive carbohydrates in the diet [5]. Increased sugar consumption leads to a shift in oral bacterial metabolism that favors the secretion of acidic byproducts. Reduced pH in the local environment results in irreversible loss of the protective enamel coating of the teeth and dysbiosis of the bacterial microbiota favoring acid-tolerant microbes [1, 5, 6]. Gingivitis is associated with inflammation of the tissues that surround and support the teeth, which results from dysbiosis and increased abundance of dental plaque forming bacteria. While the pathogenesis of caries and gingivitis are complex, poor oral hygiene is the main risk factor [7, 8].

The application of genomic detection approaches have improved our understanding of the oral microbiota and how it can reflect both health and disease [9]. In particular, the use of massively parallel bacterial 16S ribosomal RNA (rRNA) gene sequencing (16S sequencing) methods have enhanced our understanding of the oral microbiome [10, 11]. 16S sequencing typically targets one or more phylogenetically discriminating regions of the bacterial 16S rRNA gene allowing for the relative quantification of discrete bacterial taxa present in polymicrobial communities [12]. 16S sequencing is a rapidly evolving field in which technological and informatic improvements continue to increase the sensitivity and accuracy of microbiome studies [13, 14]. Multiple factors influence 16S sequencing results including sampling methods, DNA extraction, primer design, library preparation, sequencing depth, and analysis pipelines [15]. To date, there is no consensus on which parameters are ideal in the context of oral health-related disease studies.

Saliva is an easy-to-collect, inexpensive, and non-invasive biological specimen. There are different methods to collect saliva samples including with/without stimulation of saliva production among others [16]. Spontaneous or unstimulated saliva is often collected after several mouth rinses to avoid bias from recently ingested meals; however, recovery volume is often a challenge which can limit multi-omics approaches and multiple end-point assays. In contrast, stimulated saliva is also collected after mouth rinse but involves an external stimulus, such as chewing paraffin wax, which improves saliva production; however, there are concerns that saliva stimulation may result in saliva dilution and bias results [17]. There is a dearth of research on whether the saliva sampling method (stimulated vs. unstimulated) influences microbiome results. Although there are at least two prior studies that have analyzed the influence of saliva sampling method on microbiome composition, the results are conflicting. One study showed differences in taxa distribution based on the sampling method used [17], while the second study reported no differences [18]. The variability in these findings may be explained by methodological differences and small cohort size. Furthermore, no prior studies have assessed the correlation between sampling method and oral disease status.

In this study, we investigated saliva bacterial microbiome composition in the context two major oral diseases (gingivitis and dental caries) by comparing paired unstimulated and stimulated saliva samples in a cohort of 88 subjects, aged 7–18 years, collected at the time of an oral health exam.

## Materials and methods

### Study population

This study is part of a large population-based clinical study that identified participants using Medicaid files in Washington State from 2014 or 2015, enrolled them, and followed the participants longitudinally to investigate the occurrence of health outcomes in individuals with special health care needs [18]. Study participants were individuals with special health care needs aged 7–18 years registered in Medicaid. Medicaid is a public health insurance program for low-income populations. Study participants were identified as having an episodic, life-long, malignancy, or catastrophic chronic condition based on Clinical Risk Grouping methods based on Washington state Medicaid files in 2014 or 2015 [18]. Inclusion criteria were a home address located in one of three counties in Washington state: King, Pierce, or Snohomish, and a telephone number listed in the Medicaid enrollment files. For this sub-study, participants recruited between December 28, 2016, and March 4, 2017, were enrolled. Inclusion criteria included participants previously enrolled in the ongoing clinical study and consent to provide both unstimulated and stimulated saliva samples. Study participants who were unable to complete the full saliva collection procedure were excluded from this study (n = 5) [19]. In total, 88 participants were included in the analysis (S1 Fig).

**Ethics approval and consent to participate.** Informed written consent was obtained from adult participants; for minor participants, a parent or a legal caregiver provided signed consent plus verbal assent was obtained from the minor. The study was approved by Washington State Institutional Review Board and the University of Washington Institutional Review Board. No personal identifiable information was analyzed nor will be published. All participants were aware and voluntarily agreed that dental health status and microbiological data from this study will be published without any personal identifier.

### Saliva sample collection

Sequential unstimulated and stimulated saliva samples were collected from each participant using a previously published protocol developed to collect saliva from children with special health care needs [20, 21]. Participants were asked to refrain from eating or drinking for ≥2 hours prior to sample collection. Participants were seated in a passive position, and each saliva sample was collected over a 15-minute period. Immediately before sample collection, participants were asked to swallow any saliva. For the unstimulated sample, participants expectorated into a sterile 50mL tube once per minute. After a short rest period, stimulated saliva samples were collected following similar procedures except that participants were asked to chew, for 15 minutes continuously, an unflavored piece of paraffin wax, and expectorated stimulated saliva was collected once per minute on a 50mL tube [16]. Saliva samples were placed on ice and transported to a laboratory for storage at -80°C.

### Dental screening for gingivitis and untreated dental caries

After saliva collection, a dental screening was conducted by a trained and calibrated pediatric dentist or dental hygienist. The dental screening was designed to assess dental caries and gingivitis. Participants with unmet dental treatment needs were given a referral for treatment.

Dental caries were measured by visual inspection of the teeth after brushing all tooth surfaces with a dry toothbrush. Each primary and permanent tooth surface was classified as decayed, filled, or missing using the NIDCR Early Childhood Caries Collaborative Centers (EC4) Criteria [22]. Untreated dental caries were defined as the presence of at least one

decayed tooth surface. A demographic questionnaire was applied to determine variables such as sex, age, and previous antibiotic use (in the last 2 months).

Gingivitis was assessed by the presence of bleeding on probing. Using a manual UNC-15 periodontal probe, bleeding on probing was assessed at four sites (distal, buccal, mesial, and lingual) on six teeth (maxillary right first molar, maxillary right lateral incisor, maxillary left first molar, mandibular left first molar, mandibular right lateral incisor, mandibular right first molar) consistent with previously used methods [23]. Gingivitis was defined as the presence of bleeding on probing in at least 10% of the examined sites [24]. If a participant required a prophylactic antibiotic for dental exams, dental probing was not conducted.

## DNA extraction, library preparation, and sequencing

Saliva samples were thawed on ice and 1.0 ml of saliva was centrifuged at ~16,000 x g for 2 minutes. Pellets were resuspended, and DNA was extracted using DNeasy PowerSoil DNA kit (Qiagen) following the manufacturer protocol. After DNA isolation, libraries for next-generation sequencing were prepared using Quick-16S™ NGS Library Prep Kit (Zymo Research Corp). 20ng of DNA per sample was amplified using the 16S V1-V2 primer set (Zymo Research Corp.) due to high sensitivity and specificity in respiratory samples [25], and the following cycling conditions: 95˚C x 10 min., 95˚C x 30 sec., 55˚C x 30 sec., and 72˚C x 3 min. (20 cycles). To verify that sufficient amplification occurred, a fluorescence threshold was set based on manufacturer guidelines (~500,000 fluorescence threshold). Following amplification, samples were barcoded using Zymo Indexes primers ZA5xx and ZA7xx. Barcoding occurred using the following cycling parameters: 95˚C x 10min., 95˚C x 30sec., 55˚C x 30sec., and 72˚C x 3 min. (5 cycles). Final quality control checks for amplification, and sample pooling were performed using manufacturer recommendations. Pooled libraries were sequenced on an Illumina MiSeq instrument analyzing 300x2 paired-end 100.000 reads at the University of North Carolina at Chapel Hill High-Throughput Sequencing Facility. Sequence data are available from the NCBI Sequence Read Archive (accession code PRJNA1072698), in addition, individual samples metadata is publicly available at the Carolina Digital Repository (https://cdr.lib.unc.edu/) under the name of "Saliva Stimulation and microbiome project".

## Microbiome sequencing analyses

Sequence analysis was performed using the UNC Longleaf informatic environment and methods previously described [26]. Specifically, the DADA2 module and QIIME2 v.2022.10 was used to merge and denoise sequences and generate amplicon sequence variants (ASVs) as microbiome read-outs. Taxa were assigned based on matching ASVs with at least 95% sequence similarity as compared to the to the Human Oral Microbiome database (HOMD) v.15.23 reference [27] using the QIIME2 taxa classifier. Further analyses including alpha- and beta-diversity metrics were performed using QIIME2 tools. Taxa abundances were transformed to centered log ratio (CLR) and horizontal bar plots were used to show taxa enriched on each condition as compared to the average abundance in both conditions. Analysis of compositions of microbiomes with bias correction (ANCOM BC-2 analysis) [27] was performed to determine the statistical differential abundance of taxa among our comparison groups (annotated with asterisks in the differential abundance bar plots).

## Data curation and statistical analyses

A total of 176 paired samples were collected (88 stimulated saliva and 88 unstimulated) from 88 participants. As part of our quality checks, rarefaction curves were plotted (S2 Fig) to verify that stimulated and unstimulated samples reach similar sequencing depth. Four samples (2

stimulated saliva and 2 unstimulated saliva from different subjects) that contained less than 20,000 reads were eliminated because of low sequencing depth. Decontamination was performed by removing all ASVs present in the negative library preps controls (87 out of 12,333 ASVs). After quality control, a total of 86 stimulated and 86 unstimulated saliva samples were included in the microbial diversity analyses. For the disease-based analyses, a "healthy group" was created (n = 37) representing participants who had no untreated caries and no gingivitis, to avoid bias when comparing each disease group (participants who had either untreated dental caries or gingivitis) with the healthy group. Faith's phylogenetic alpha diversity (Faith's PD) and principal components PERMANOVA analyses (beta diversity Bray Curtis) were used to initially assess microbial diversity among groups. Beta diversity ANCOM metrics were used to determine relative abundance of taxa between saliva sample types and diseases groups. In addition, microbial diversity among demographic and clinical factors were also explored to identify possible confounding factors. Additionally, Faith's PD and Bray Curtis microbial composition distance analysis were used to determine pairwise differences within subjects (stimulated vs. unstimulated) as compared to across subjects, and whether pairwise differences by subjects were also informative of the diversity differences driven by oral diseases. Data outputs were generated in python and relabeled and sized in Adobe Illustrator for presentation.

## Results and discussion

### Study population

A total of 88 participants were included in this study. Demographic and relevant clinical features of the cohort are summarized in Table 1. Forty-two percent were female, 6.8% Black or African American, 35.2% White, 11.4% American Indian or Alaska Native, 6.8% Asian or Pacific Islander, and 35.2% reported as Multiracial. Thirty-five percent also self-identified as Hispanic. Approximately 16% of participants had used antibiotics within two months of saliva collection. At the time of sampling, 17% were found to have untreated caries and 35.3% had gingivitis.

**Table 1. Clinical/demographic characteristics of the study population.**

| Item | n |
|---|---|
| Total subjects, no. | 88 |
| Female sex (%) | 37 (42%) |
| Age, average (standard deviation) | 12.4 (3.2) |
| Race, no. (%) | |
| Black or African American | 6 (6.8%) |
| White | 31 (35.2%) |
| American Indian or Alaska Native | 10 (11.4%) |
| Asian or Pacific Islander | 6 (6.8%) |
| Other or Multiracial | 31 (35.2%) |
| Not reported | 4 (4.5%) |
| Hispanic, no. (%) | 31 (35.2%) |
| Antibiotic use (<2 months), no. (%) | 14 (15.9%) |
| Oral health* | |
| Untreated dental caries, no. (%) | 15 (17.0%) |
| Gingivitis, no. (%)** | 31 (35.3%) |
| Healthy controls, no. (%) | 37 (42.0%) |

\* 2 subjects had untreated dental caries and gingivitis

\*\* 9 missing values

As a first step, we analyzed antibiotic use (in the past 2 months), sex, and age as variables that could impact oral bacterial microbiome metrics; however, none were significantly associated with microbial composition in both stimulated and unstimulated saliva samples (S3–S5 Figs).

## Saliva stimulation is associated with differences in oral microbiome composition

We evaluated the association between saliva sampling method (stimulated vs unstimulated) and bacterial microbiome composition across the study population. Our data indicates a statistically significant difference (p<0.01) in alpha diversity (Faith's PD) between the average taxa composition in unstimulated versus stimulated saliva samples with stimulated saliva showing greater diversity (Fig 1A). Taxonomic beta diversity analysis at the genus level and

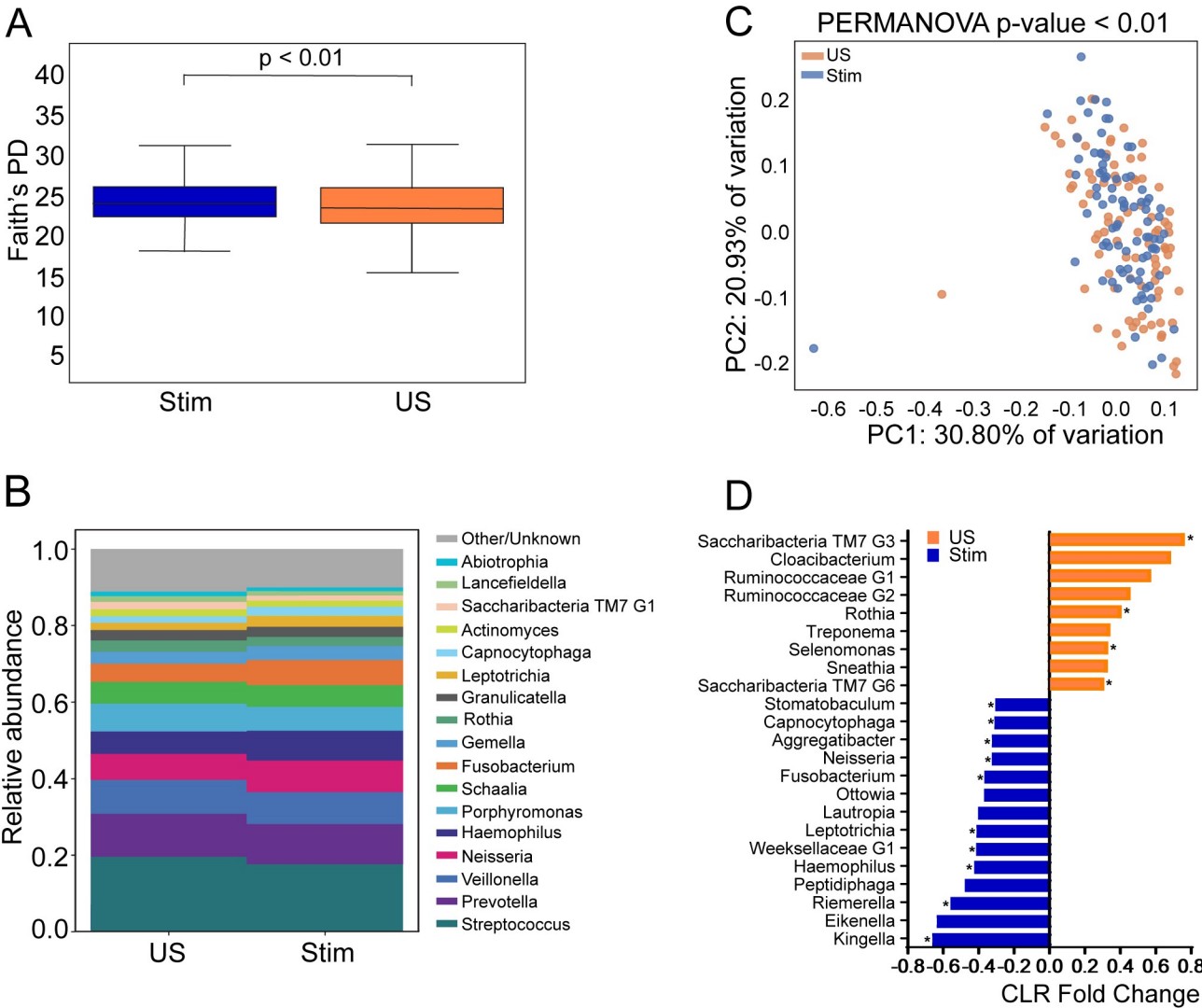

**Fig 1. Diversity metrics comparing microbiome composition using stimulated (Stim) versus unstimulated (US) saliva.** A) Alpha diversity (Faith's PD) comparison of Stim and US showing statistically significant differences in microbial composition. B) Relative abundance of taxa at genus level comparing Stim versus US saliva. C) Beta diversity (Bray Curtis) Principal components plot comparing Stim vs US saliva by PERMANOVA statistical test. D) Beta diversity centered log ratio (CLR) fold change between Stim and US at genus taxonomic level (Asterisk mark ANCOM statistically significant differences, p<0.05).

PERMANOVA Bray Curtis revealed differential abundance of taxa in stimulated saliva compared to unstimulated saliva samples (Fig 1B and 1C). Specifically, we found several taxa, such as *Kingella*, *Riemerella*, *Haemophilus*, among others that were significantly more abundant in stimulated saliva. In contrast, *TM7*, *Rothia*, *Selenomas* among others were significantly higher in unstimulated saliva (Fig 1D). As expected, we found that the difference in microbial composition within subjects (paired stimulated versus unstimulated samples) was significantly less than that between subjects (p<0.01) (S6A Fig).

## Bacterial taxa related to gingivitis status are enriched only in stimulated saliva

We determined whether the saliva sampling method reflects changes in the bacterial microbiome composition in subjects with gingivitis. Initial alpha diversity metrics (Faith's PD) indicate differences between the saliva bacterial microbiome in participants with gingivitis compared to healthy controls (without gingivitis and without untreated dental caries), however, the differences were only statistically significant in stimulated saliva (p<0.01) but not in unstimulated saliva (p = 0.12) (Fig 2A and 2B). Additional analysis using beta diversity metrics (PERMANOVA-Bray Curtis) revealed different taxa distribution in the study group with gingivitis as compared to healthy controls (Fig 2C and 2D, also S7A and S7B Fig); however, this difference was only statistically significant in stimulated saliva (p<0.05). We observed that

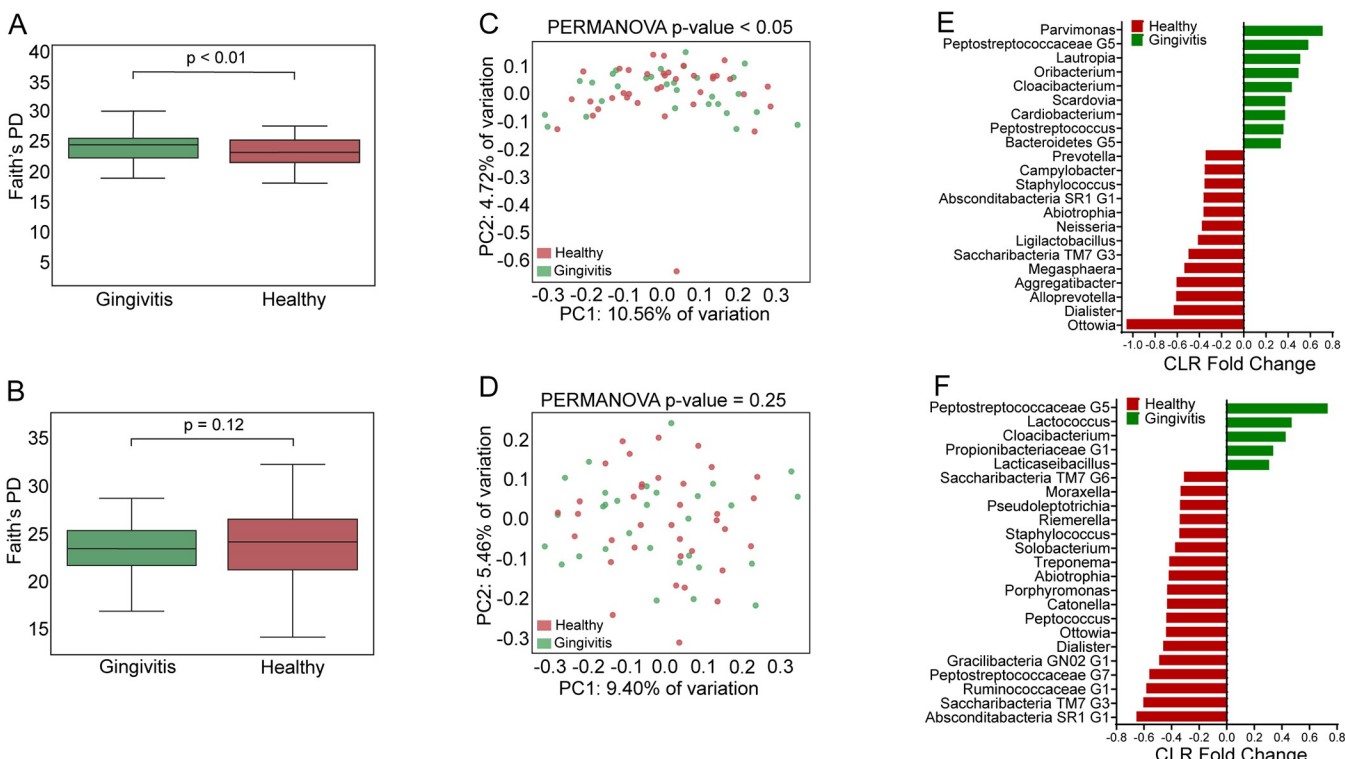

**Fig 2. Comparison of microbiome composition between participants who had gingivitis versus healthy controls (no caries, no gingivitis) using either stimulated (Stim) or unstimulated (US) saliva.** Alpha diversity (Faith's PD) comparison of microbiome in Stim (A) and US (B) showing statistically significant different diversity only in Stim (p<0.05) but not on US. Principal components plot comparing subjects with untreated gingivitis versus healthy controls in both Stim (C) and US (D) saliva; PERMANOVA beta diversity analysis indicates significantly different taxonomic composition (p<0.05) in Stim based on gingivitis status. Beta diversity centered log ratio (CLR) fold change showing differential enrichment of taxa at the genus level comparing subjects with gingivitis (yes) and healthy controls (no) in both Stim (E) and US (F) samples.

taxa previously associated with gingivitis including *Peptostreptococcus* [28], were enriched in the group of subjects with gingivitis only when assessed using stimulated saliva. Faith's PD alpha diversity differences between paired stimulated and unstimulated samples for each subject (ΔFaith's PD) and Bray Curtis distance (Paired Beta diversity) were not significantly different (p>0.05) between gingivitis affected subjects and healthy controls (S6B and S6D Fig).

## Oral microbiome differences were observed with untreated dental caries in stimulated saliva

We next examined whether saliva collection method is associated with differences in bacterial microbiome composition between study participants with untreated caries and healthy controls. Our analysis showed no difference in alpha diversity, regardless of saliva collection method (Fig 3A and 3B). However, microbial community composition differed significantly in saliva between subjects with untreated caries compared to healthy controls by PERMANOVA beta diversity analysis (Fig 3C and 3D, also S7C and S7D Fig). While the composition of the saliva bacterial microbiome (beta diversity) was significantly different between untreated caries and healthy control groups in both saliva sample types, the number of taxa showing differential relative abundance at the genus level was substantially greater in stimulated saliva (Fig 3E and 3F). Additionally, taxa previously reported to be associated with dental caries including *Capnocytophaga* [29], *and Aggregatibacter* [30], were enriched in subjects with untreated dental caries in stimulated saliva but not in unstimulated saliva samples. Finally, the untreated caries group was further compared to healthy controls using Faith's PD alpha

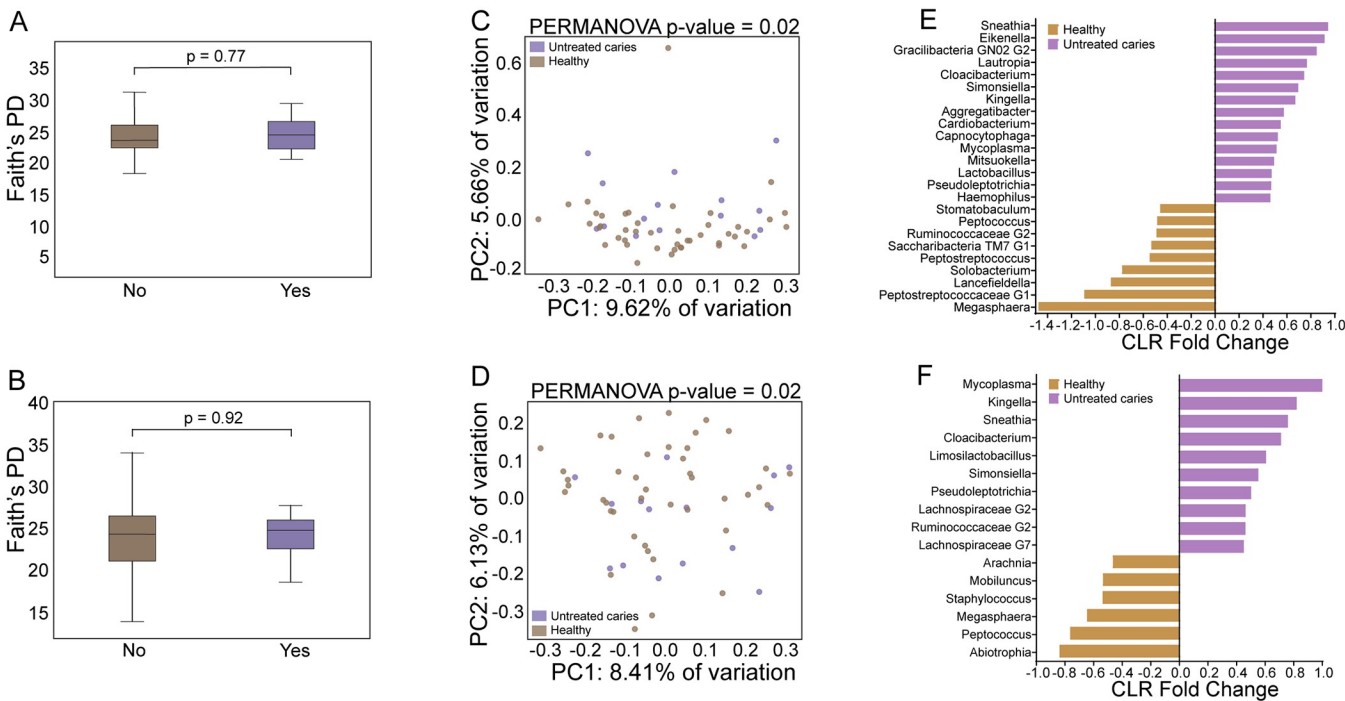

**Fig 3. Comparison of microbiome composition between participants with untreated caries versus healthy controls (no caries, no gingivitis) using either stimulated (Stim) or unstimulated (US) saliva.** Alpha (Faith's PD) diversity comparison of microbiome based on untreated caries status in Stim (A) and US saliva (B). Principal components plot comparing subjects with untreated dental caries Healthy controls in both Stim (C) and US saliva (D); PERMANOVA beta diversity analysis indicates significantly different taxonomic composition based on caries status. Beta diversity centered log ratio (CLR) fold change depicting differential enrichment of taxa at the genus level comparing subjects with untreated caries (yes) and healthy controls (no) in both Stim (E) and US (F) samples.

diversity differences between stimulated and unstimulated for each subject (ΔFaith's PD, S6C Fig) and Bray Curtis distance (Paired Beta diversity, S6E Fig); however, no significant differences were found (p>0.05).

## Discussion

In the present study, we found that saliva sampling method (unstimulated versus stimulated saliva) is associated with differences in oral bacterial microbiome metrics in general and in the context of oral diseases (gingivitis and untreated dental caries). Our main findings indicate that unstimulated and stimulated saliva show significant differences in taxa composition and that the influence of oral diseases in the bacterial microbiome composition is dependent on the sampling method used. Furthermore, several taxa known to be associated with dental caries or gingivitis were only enriched in stimulated saliva in our study.

There are multiple sample types used to measure oral microbiome composition such as supragingival plaque, subgingival/submucosal plaque, infected root canals, mucosal surfaces, and saliva [31–33]. The collection of many of these sample types is invasive and/or requires a trained healthcare provider and dedicated equipment. However, saliva collection is non-invasive, does not require special training and can be collected at home, preserved and sent to researchers to measure the oral microbiome composition and microbial changes that accompany diseases like dental caries and gingivitis [34–36]. Saliva collection can be performed using different methods including unstimulated and stimulated saliva. Stimulated saliva allows the collection of a larger sample volume in a shorter amount of time [37]. Our results suggest that stimulated saliva provides a better representation of taxa associated with common oral diseases, which may be explained by the fact that mechanical stimulation (chewing paraffin wax) not only increases saliva production but also could cause the displacement of taxa present on teeth and in periodontal pockets [38].

Although previous reports have found major effects of antibiotic use on oral microbiome composition [39], we did not see a similar outcome in our study. This may be explained by differences in the timing of antibiotic use relative to saliva sample collection. Previous studies have shown that the saliva microbiome is resilient and returns to baseline quickly after antibiotic cessation [40]. Subjects in this study were not on antibiotics at the time of sample collection and were only asked to report if they had used antibiotics in the prior 2 months.

Our study demonstrates that there are significant differences in oral microbiome composition by comparing average unstimulated versus average stimulated saliva from the same participants. Additionally, our results indicate the use of stimulated saliva provides more robust detection of taxa-related oral health status. Specifically, taxa previously associated with gingivitis are better represented in stimulated saliva, which suggests that the collection of stimulated saliva for oral microbiome studies allows for a more complete assessment of microbes than unstimulated saliva. Although no significant differences in alpha or beta diversity were found comparing disease groups versus healthy controls using pairwise differences of unstimulated versus stimulated paired by subjects, we do see significant differences by comparing the average microbial composition between stimulated as compared to unstimulated. This may be explained by the reduction in statistical power in the paired comparison given that 4 subjects only had one type of sample (2 stimulated only and 2 unstimulated only).

Our study has several potential limitations including 1) a small number of participants with untreated dental caries, reduced our which statistical power, 2) the study population had special health care needs which could limit the generalizability of our findings to the larger community, 3) the use of V1-V2 primers for library preparation may decrease taxa representation beyond commonly found taxa in oral microbiome, 4) the order of sample collection

(unstimulated followed by stimulated) may have altered the composition of the stimulated saliva; however, we observed that microbial diversity increases in stimulated samples (Fig 1), and 5) the age group of our study population limit the generalizability of the findings to older and younger individuals.

Although larger prospective studies may be needed to further validate our findings, given the current shift to at home testing, stimulated saliva could be useful for health diagnosis, research, and epidemiological surveillance of oral health in rural, remote, and at-risk populations.

## Conclusions

In summary, we found that the use of unstimulated or stimulated saliva to determine oral microbiome composition provides significantly different results. We demonstrated that oral microbiome metrics are significantly different when comparing oral disease groups (untreated dental caries and gingivitis). In addition, taxa previously known to be associated with caries or gingivitis are specifically enriched using stimulated saliva in participants diagnosed with the respective oral disease. Our data indicate that stimulated saliva provides a better representation of oral health-related taxa compared to unstimulated saliva and that saliva sampling method is an important consideration in study designs.

## Supporting information

**S1 Fig. Flow diagram explaining the selection criteria of the study population.** After the selection based on our inclusion and exclusion criteria, 88 participants provided stimulated and unstimulated saliva samples.
(PDF)

**S2 Fig. Rarefaction plot showing taxonomic units observed by sequencing depth in stimulated (Stim) and unstimulated (US) saliva samples.**
(PDF)

**S3 Fig. Comparison of microbiome composition between participants among reported antibiotic use in the past 2 months using either stimulated (Stim) or unstimulated saliva (US).** Alpha diversity (Faith's PD) comparison of microbiome in Stim (A) and US (B). Principal components plot and PERMANOVA beta diversity analysis comparing Stim (C) and US (D) among participants with and without reported antibiotic use. Beta diversity centered log ratio (CLR) fold change showing differential enrichment of taxa at the genus level between Stim (E) and US (F).
(PDF)

**S4 Fig. Comparison of microbiome composition between participants among reported sex using either stimulated (Stim) or unstimulated saliva (US).** Alpha diversity (Faith's PD) comparison of microbial composition in Stim (A) and US (B). Principal components plot comparing Stim (C) and US (D) based on reported sex, PERMANOVA beta diversity test. Beta diversity centered log ratio (CLR) fold change showing differential enrichment of taxa at the genus level between Stim (E) and US (F).
(PDF)

**S5 Fig. Plot showing correlation between alpha diversity (Faith's PD) and age.** No significant differences were found using Spearman correlation (p = 0.67).
(PDF)

**S6 Fig. Paired analysis of microbial composition within and across samples by disease status.** A) Bray Curtis Beta diversity distance of microbial composition within subjects (paired) as compared to across subjects (unpaired). Pairwise Faith's PD difference of microbial composition between stimulated and unstimulated samples by gingivitis status (B) and untreated caries (C). Pairwise Bray Curtis Beta diversity distance difference of microbial composition between stimulated and unstimulated samples by gingivitis status (D) and untreated caries (E). (PDF)

**S7 Fig.** Principal components plots using PC1 and PC3 comparing subjects with gingivitis versus healthy controls in both Stim (A) and US (B) saliva, and plots comparing subjects with untreated caries versus healthy controls in both Stim (C) and US (D) saliva. PERMANOVA beta diversity analysis indicates significant difference (p<0.05) in taxonomic composition. (PDF)

## Acknowledgments

We acknowledge and thank all research subjects and health personnel for their contributions to this study.

## Author Contributions

**Conceptualization:** Cristian Roca, Alaa A. Alkhateeb, Jade K. Macdonald, Donald L. Chi, Matthew C. Wolfgang.

**Data curation:** Cristian Roca, Alaa A. Alkhateeb, Bryson K. Deanhardt, Donald L. Chi, Jeremy R. Wang, Matthew C. Wolfgang.

**Formal analysis:** Cristian Roca, Jeremy R. Wang, Matthew C. Wolfgang.

**Funding acquisition:** Donald L. Chi, Jeremy R. Wang, Matthew C. Wolfgang.

**Investigation:** Cristian Roca, Alaa A. Alkhateeb, Bryson K. Deanhardt, Jade K. Macdonald, Donald L. Chi, Jeremy R. Wang, Matthew C. Wolfgang.

**Methodology:** Cristian Roca, Alaa A. Alkhateeb, Bryson K. Deanhardt, Jade K. Macdonald, Donald L. Chi, Jeremy R. Wang, Matthew C. Wolfgang.

**Project administration:** Cristian Roca, Alaa A. Alkhateeb, Donald L. Chi, Jeremy R. Wang, Matthew C. Wolfgang.

**Resources:** Cristian Roca, Bryson K. Deanhardt, Jade K. Macdonald, Donald L. Chi, Jeremy R. Wang, Matthew C. Wolfgang.

**Software:** Cristian Roca, Jeremy R. Wang, Matthew C. Wolfgang.

**Supervision:** Cristian Roca, Alaa A. Alkhateeb, Bryson K. Deanhardt, Jade K. Macdonald, Donald L. Chi, Jeremy R. Wang, Matthew C. Wolfgang.

**Validation:** Cristian Roca, Bryson K. Deanhardt, Donald L. Chi, Jeremy R. Wang, Matthew C. Wolfgang.

**Visualization:** Cristian Roca, Donald L. Chi, Jeremy R. Wang, Matthew C. Wolfgang.

**Writing – original draft:** Cristian Roca, Alaa A. Alkhateeb, Bryson K. Deanhardt, Jade K. Macdonald, Donald L. Chi, Jeremy R. Wang, Matthew C. Wolfgang.

**Writing – review & editing:** Cristian Roca, Alaa A. Alkhateeb, Bryson K. Deanhardt, Jade K. Macdonald, Donald L. Chi, Jeremy R. Wang, Matthew C. Wolfgang.

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
