## [Decision Letter · Decision Letter 0]

29 Jan 2024

PONE-D-23-40989Saliva sampling method influences oral microbiome composition and taxa distribution associated with oral diseasesPLOS ONE

Dear Dr. Wolfgang,

Thank you for submitting your manuscript to PLOS ONE. After careful consideration, we feel that it has merit but does not fully meet PLOS ONE’s publication criteria as it currently stands. Therefore, we invite you to submit a revised version of the manuscript that addresses the points raised during the review process.

 Please submit your revised manuscript by Mar 14 2024 11:59PM. If you will need more time than this to complete your revisions, please reply to this message or contact the journal office at plosone@plos.org. Please include the following items when submitting your revised manuscript:A rebuttal letter that responds to each point raised by the academic editor and reviewer(s). You should upload this letter as a separate file labeled 'Response to Reviewers'.A marked-up copy of your manuscript that highlights changes made to the original version. You should upload this as a separate file labeled 'Revised Manuscript with Track Changes'.An unmarked version of your revised paper without tracked changes. You should upload this as a separate file labeled 'Manuscript'.If applicable, we recommend that you deposit your laboratory protocols in protocols.io to enhance the reproducibility of your results. Protocols.io assigns your protocol its own identifier (DOI) so that it can be cited independently in the future. For instructions see: https://journals.plos.org/plosone/s/submission-guidelines#loc-laboratory-protocols. Additionally, PLOS ONE offers an option for publishing peer-reviewed Lab Protocol articles, which describe protocols hosted on protocols.io. Read more information on sharing protocols at https://plos.org/protocols?utm_medium=editorial-email&utm_source=authorletters&utm_campaign=protocols.

We look forward to receiving your revised manuscript.

Kind regards,

Farah Al-Marzooq, MD, PhD

Academic Editor

PLOS ONE

2. Please provide additional details regarding participant consent. In the ethics statement in the Methods and online submission information, please ensure that you have specified what type you obtained (for instance, written or verbal, and if verbal, how it was documented and witnessed).

“C.R., D.L.C and M.C.W. were funded in part by support from the National Institute of Dental and Craniofacial Research, NIH (U01DE030418), and J.R.W. was funded by NIH K01 DK119582.”

5. We notice that your supplementary figures are uploaded with the file type 'Figure'. Please amend the file type to 'Supporting Information'. Please ensure that each Supporting Information file has a legend listed in the manuscript after the references list.

Reviewers' comments:

Reviewer's Responses to Questions

**Comments to the Author**

1. Is the manuscript technically sound, and do the data support the conclusions?

Reviewer #1: Partly

2. Has the statistical analysis been performed appropriately and rigorously? 

Reviewer #1: Yes

3. Have the authors made all data underlying the findings in their manuscript fully available?

Reviewer #1: No

4. Is the manuscript presented in an intelligible fashion and written in standard English?

Reviewer #1: Yes

5. Review Comments to the Author

Reviewer #1: The manuscript titled 'Saliva Sampling Method Influences Oral Microbiome Composition and Taxa Distribution Associated with Oral Diseases,' submitted to PLOS ONE, compares the oral microbiome composition using simulated and unstimulated saliva sampling methods. In general, this study is well-conducted, and the figures are good. However, there are many areas that require attention before it can be accepted for publication.

Major comments:

1. Considering the significance of stimulated and unstimulated saliva sampling methods for this paper, the authors are required to provide some background on these methods in the introduction.

2. Including a figure illustrating the study design can enhance readers' understanding of the study.

3. The authors are required to provide the accession number for the deposited raw reads.

Minor comments:

1. The writing for stimulated saliva sampling collection should be clearer in methods.

2. In lines 155-156, where it is mentioned, 'Taxa were assigned based on matching ASVs to the Human Oral Microbiome database v.15.23 (27),' the authors are advised to provide the criteria used for taxa classification.

3. The authors mentioned ANCOM analysis in line 160; they are advised to provide the full name of it and include a citation.

4. The authors used PC1 and PC3 for PC plots, whereas it is more common to use PC1 and PC2 for such plots. It is advised to consider a three-dimensional figure if the authors wish to retain PC3.

6. PLOS authors have the option to publish the peer review history of their article (what does this mean?). If published, this will include your full peer review and any attached files.

Reviewer #1: No

---

## [Author Response · Author response to Decision Letter 0]

15 Feb 2024

We would like to thank the editor and reviewer for their insightful comments and valuable suggestions. We have addressed all comments and believe that it has resulted in a more concise and improved manuscript. 

The manuscript has been re-organized to comply with the journal requirements for the main text and the authors affiliations.

2. Please provide additional details regarding participant consent. In the ethics statement in the Methods and online submission information, please ensure that you have specified what type you obtained (for instance, written or verbal, and if verbal, how it was documented and witnessed).

Additional details on the ethics statement were added in the Methods so that it clarifies whether a written or verbal consent/assent was obtained from the study participants (lines 108-115). 

“C.R., D.L.C and M.C.W. were funded in part by support from the National Institute of Dental and Craniofacial Research, NIH (U01DE030418), and J.R.W. was funded by NIH K01 DK119582.” Please state what role the funders took in the study. If the funders had no role, please state: "The funders had no role in study design, data collection and analysis, decision to publish, or preparation of the manuscript." If this statement is not correct you must amend it as needed.

A statement about the role of the funders of the study was added to the author’s contribution section (lines 349-351) and the cover letter.

The ethics statement was moved to the Materials and Methods section and deleted from its previous position (lines 108-115).

5. We notice that your supplementary figures are uploaded with the file type 'Figure'. Please amend the file type to 'Supporting Information'. Please ensure that each Supporting Information file has a legend listed in the manuscript after the references list.

The file type was corrected, and a legend of each supporting information figure is now present in the manuscript after references (lines 472-504).

References have been revised. Some references have been added to comply with the reviewer’s comments.

Reviewer #1 (Comments for the Author):

Reviewer #1: The manuscript titled 'Saliva Sampling Method Influences Oral Microbiome Composition and Taxa Distribution Associated with Oral Diseases,' submitted to PLOS ONE, compares the oral microbiome composition using simulated and unstimulated saliva sampling methods. In general, this study is wellconducted, and the figures are good. However, there are many areas that require attention before it can be accepted for publication.

Major comments:

1. Considering the significance of stimulated and unstimulated saliva sampling methods for this paper, the authors are required to provide some background on these methods in the introduction.

The introduction has been amended to now include background information on stimulated and unstimulated saliva samples (lines 70-78).

2. Including a figure illustrating the study design can enhance readers' understanding of the study.

A flow diagram has been added as Figure S1 in the supporting information to clarify the participants selection and study design.

3. The authors are required to provide the accession number for the deposited raw reads.

The accession number for the raw sequences and the metadata have been added to the text (lines 334-338).

Minor comments:

1. The writing for stimulated saliva sampling collection should be clearer in methods. 

A clarification for the stimulated saliva collection was included in the methods section (lines 124-127).

2. In lines 155-156, where it is mentioned, 'Taxa were assigned based on matching ASVs to the Human Oral Microbiome database v.15.23 (27),' the authors are advised to provide the criteria used for taxa classification.

Details related to the taxonomic classification were added to the text (lines 172-174).

3. The authors mentioned ANCOM analysis in line 160; they are advised to provide the full name of it and include a citation.

The full name for ANCOM and the respective citation was added to the text (lines 178-179).

4. The authors used PC1 and PC3 for PC plots, whereas it is more common to use PC1 and PC2 for such

plots. It is advised to consider a three-dimensional figure if the authors wish to retain PC3.

Regarding Figure 1, there was a typographical error, and it already presented PC1 and PC2. The error was corrected. All remaining figures were revised to show PC1 and PC2. In addition, the plots for PC1 and PC3 were included as a supplementary figure (Fig. S7).

---

## [Editor Report · Decision Letter 1]

11 Mar 2024

Saliva sampling method influences oral microbiome composition and taxa distribution associated with oral diseases

PONE-D-23-40989R1

Dear Dr. Wolfgang,

We’re pleased to inform you that your manuscript has been judged scientifically suitable for publication and will be formally accepted for publication once it meets all outstanding technical requirements.

Kind regards,

Farah Al-Marzooq, MD, PhD

Academic Editor

PLOS ONE
---

## [Editor Report · Acceptance letter]

20 Mar 2024

PONE-D-23-40989R1 

PLOS ONE

Dear Dr. Wolfgang, 

I'm pleased to inform you that your manuscript has been deemed suitable for publication in PLOS ONE. Congratulations! Your manuscript is now being handed over to our production team.

Kind regards, 

on behalf of

Dr. Farah Al-Marzooq 

Academic Editor

PLOS ONE